# Estimation of 6D Object Pose Using a 2D Bounding Box

**DOI:** 10.3390/s21092939

**Published:** 2021-04-22

**Authors:** Yong Hong, Jin Liu, Zahid Jahangir, Sheng He, Qing Zhang

**Affiliations:** 1State Key Laboratory of Information Engineering in Surveying, Mapping and Remote Sensing, Wuhan University, Wuhan 430079, China; 2019106190017@whu.edu.cn (Y.H.); Z.jahangir93@whu.edu.cn (Z.J.); 2014301610342@whu.edu.cn (S.H.); 2College of Intelligent Systems Science and Engineering, Harbin Engineering University, Harbin 150001, China; zhq402@hrbeu.edu.cn

**Keywords:** 6D pose estimation, quaternion, Bounding Box Equation, LineMod

## Abstract

This paper provides an efficient way of addressing the problem of detecting or estimating the 6-Dimensional (6D) pose of objects from an RGB image. A quaternion is used to define an object′s three-dimensional pose, but the pose represented by q and the pose represented by -q are equivalent, and the L2 loss between them is very large. Therefore, we define a new quaternion pose loss function to solve this problem. Based on this, we designed a new convolutional neural network named Q-Net to estimate an object’s pose. Considering that the quaternion′s output is a unit vector, a normalization layer is added in Q-Net to hold the output of pose on a four-dimensional unit sphere. We propose a new algorithm, called the Bounding Box Equation, to obtain 3D translation quickly and effectively from 2D bounding boxes. The algorithm uses an entirely new way of assessing the 3D rotation (R) and 3D translation rotation (t) in only one RGB image. This method can upgrade any traditional 2D-box prediction algorithm to a 3D prediction model. We evaluated our model using the LineMod dataset, and experiments have shown that our methodology is more acceptable and efficient in terms of L2 loss and computational time.

## 1. Introduction

Object detection and localization have always been hot computer vision topics. Conventional methods, such as you only look once (YOLO) [1], single-shot detector (SSD) [2], etc., have demonstrated impressive performance in the 2D region. The same semantic representation of the objective 3D world, on the other hand, is more difficult to achieve due to a lack of information about the object’s rotation and position in relation to the camera. An object pose has 6 degrees of freedom (6DOF), 3 degrees of freedom in translation, and 3 degrees in rotation, and this information is needed in several applications for robotic and scene interpretation [3]. The measurement of 6D poses has been previously the subject of extensive study. Consequently, estimating an object’s 6D pose is challenging because of different elements, e.g., shape varieties and their different angles of visualization, illumination, and occlusions among objects.

Feature-based and template-based techniques are the most traditional approaches in this area. A feature-based algorithm extracts local features from points of interest in the image with local descriptors to align them with 3D models to retrieve 6D poses [4,5,6,7,8]. For example, [8] used Scale Invariant Feature Transform (SIFT) descriptors and clustered images in a single model from similar perspectives [7], which provided a fast and flexible sensing approach to detect and estimate objects. However, these methods suffer from a common limitation in that they require sufficient texture on the objects. Feature learning methods [9,10] were proposed to perform better matching perspectives in dealing with inadequate texture artifacts. Feature-based techniques compare the characteristic points between 3D models and images. Nevertheless, such methods are functional only where there are rich textures on the objects. They are unable to manage artifacts that are texture-less [11]. Further, their basic design is time-consuming and multi-stage. A rigid template was examined in the image in a template-based way [12,13], and distance measurement was measured to better fit. These approaches can work precisely and rapidly but do not work properly if clutter and occlusions are involved.

Depth cameras make RGB-D object pose estimation methods more prevalent [14,15,16,17,18,19,20,21]. For instance, an algorithm suitable for textured and texture-less generic objects was studied in [19]. Another study was conducted based on multiple views to identify an object’s pose with 6DOF for multiple object cases in a crowded scene [20]. For RGB-D images, another study established a flexible and fast process [14]. The task can be made more accessible by using depth images, but this requires extra hardware for acquiring depth data.

Approaches focused on convolutional neural networks (CNNs) have become mainstream in recent years to solve problems related to 6D posing, particularly camera poses [22,23] and objects [24,25,26,27,28]. Both [22] and [23] trained CNNs to return the 6D camera pose directly. The prediction of camera poses is much simpler since there is no necessity to identify any object. A study used quaternion as rotation representation but omitted the spherical constraint, using an unreasonable loss function [22]. The problem was tackled through more scientific loss function growth [23]. Another study [25] used CNNs for direct 3D object regression, but their results were limited to 3D rotation estimates only, with no regard for 3D translation. The 6D pose estimation has been expanded by the SSD recognition method [2]. The authors of [26] have transformed pose recognition into two simple steps, classification of angles and in-plane rotation. However, incorrect classification can result in an inaccurate pose estimate at either point. A segmentation network was used to locate objects in another analysis, called BB8, for eight corners of the bounding box [27]. To calculate 2D projections of the corners of the 3D boundary box around the object, another CNN was then used. A Perspective-n-Point (PnP) algorithm determined the 6D pose. The CNN was eventually conditioned to improve the pose. BB8 is highly reliable but time-consuming. Another study was conducted [24] that advanced the YOLO object detection framework [1] to anticipate two-dimensional projections at the 3D bounding box corners. The excessive amount of 2D points were regressed, thus increased the learning problem and reduced the speed of learning [24,27]. Contrary to the above, a CNN algorithm proposed regressing the orientation of a 3D object and then combining those estimates with geometrical strictures generated by a 2D object bounding box to produce a complete 3D bounding box [28], although this approach usually requires 4096 linear equations to be solved. In extraordinary cases, e.g., the KITTI dataset [29], a project of Karlsruhe Institute of Technology and Toyota Technology Institute at Chicago, the pitch and roll angle of artifacts are both zero, and 64 equations still have to be solved, which makes the method computationally expensive. The above-mentioned issues are avoided in our method. A study based on deep learning proposed a novel method to localize human actions in videos spatio-temporally with integrating an optical flow subnet. The designed new architecture is able to perform action localization and optical flow estimation jointly in an end-to-end manner [30].

Deep learning models have recently become commonplace in estimating the pose of a 6D object. We proposed a generic framework in this paper that overcomes the limitations of previous methods of estimating 6D objects. We have developed an entirely new way of assessing the 3D rotation *R* and 3D translation rotation **t** in only one RGB image. The 6D Pose estimation procedure was divided into two main phases. In the first step, the 3D rotation *R* was regressed by a new convolutional neural network we named Q-Net. In step two, the Bounding Box Equation algorithm included the 3D translation **t** generated with *R* and the 2D bounding box on the original image. 

We evaluated our approach using the LineMod dataset method [13], a 6D pose prediction reference. Our experiments have shown that the Q-Net and Bounding Box Equation worked effectively and fruitfully on these demanding data sets, producing cutting-edge results irrespective of the complicated scenes on the images. We also used our system to identify the 6D pose of ordinary objects in everyday life, and the results were also very satisfactory. The objective world is three-dimensional, and as such, in practical applications we are more concerned about the three-dimensional position and posture of the target relative to the camera, which will be more convenient for us to understand the target’s semantic information in the scene. In short, our job has the aforementioned benefits and accomplishments:Our approach requires no depth information and works on textured and texture-free images. It also has practical significance and can be used in everyday life.Compared to previous 2D detection systems, our approach worked robustly and more easily.We designed a channel normalize layer for uniting n-dimensional feature vectors formed by neurons in n feature maps. The calculation layer can directly output the attitude quaternion of the target. At the same time, we defined a new quaternion pose loss function for measuring the similarity of pose between models to predict and ground truth.We implement the Bounding Box Equation, a new algorithm for the effective and accurate 3D translation using the R and 2D bounding box.

The rest of the paper is structured accordingly. In Section 2.2., we introduced the network architecture, and Section 2.2 and Section 2.3 introduced a new network model Q-net for predicting attitude quaternion. Section 2.4. contains the loss function for regressing the quaternion of the target attitude. Section 2.5. studies the backpropagation algorithm for quaternion calculation. Section 2.6. holds the prediction of object position through the BB3D algorithm. Experimental results and their evaluations are elaborated in Section 3. The last two sections, i.e., Section 2.4 and Section 2.5, consist of discussion and conclusion. 

## 2. Materials and Methods

### 2.1. Data

In this study, we used the LineMod data set [13] and the KITTI data set (http://www.cvlibs.net/datasets/kitti/eval_object.php?obj_benchmark=3d, accessed on 12 February 2021) to verify our new method.

### 2.2. Framework of the Study

Figure 1 depicts the method of our research. The framework was divided into few steps: (1) 2D object Detection, (2) obtaining 2D bounding box, (3) converting to 48 × 48 size, (4) building Q-Net regression for quaternion **q**, (5) converting **q** to rotation matrix **R**, (6) applying bounding box equation to find 3D translation **T**, and (7) visualizing the 3D rotation and translation by projecting the eight corners of the 3D bounding box onto the image.

### 2.3. Q-Net Architecture

The three significant depictions of 3D rotation are Euler angles, the rotation matrix, and the quaternion unit. Euler angles can be understood by defining three rotational angles along three axes. However, the regression of Euler angles is often an arduous task because of many issues. For example, poses that are visibly quite identical may be far away in space at the Euler angle [31]. The rotation matrix is an orthogonal 3 × 3-element matrix with several particular characteristics. However, it is unacceptable to regress since orthogonality is difficult to implement when learning a 3D rotation representation by backpropagation [23].

A unit quaternion is even more appropriate than the former two representations. We selected an appropriate unit quaternion, *q* = (*q*_0_, *q*_1_, *q*_2_, *q*_3_) as the description to prevent problems induced by Euler’s angles and rotation matrix. There is a very convenient conversion relationship between quaternion and attitude matrix.
(1)$$R=\left[\begin{array}{ccc} {q_{0}}^{2}+{q_{1}}^{2}-{q_{2}}^{2}-{q_{3}}^{2} & 2(q_{1}q_{2}+q_{0}q_{3}) & 2(q_{1}q_{3}-q_{0}q_{2})\\ 2(q_{1}q_{2}-q_{0}q_{3}) & {q_{0}}^{2}-{q_{1}}^{2}+{q_{2}}^{2}-{q_{3}}^{2} & 2(q_{2}q_{3}+q_{0}q_{1})\\ 2(q_{1}q_{3}+q_{0}q_{2}) & 2(q_{2}q_{3}-q_{0}q_{1}) & {q_{0}}^{2}-{q_{1}}^{2}-{q_{2}}^{2}+{q_{3}}^{2} \end{array}\right]$$

We established the CNN, Q-Net, to predict the quaternion. Figure 2 shows the architecture. 

Q-net is a kind of network (or branch network) that was used to output the target’s attitude quaternion. Q-net’s core added a vector unit calculation to its output part, which was used to output the quaternion as follows.

Unit quaternion achieved the condition of limitations: (2)$${q_{0}}^{2}+{q_{1}}^{2}+{q_{2}}^{2}+{q_{3}}^{2}=1$$

The network layer does not ensure that the network output is a unit vector. Similar to [31], we added a further layer, the *q* normalization layer, to carry the output of the unit’s sphere. The forward propagation in the q normalization layer is given as:(3)$$q_{i}=\frac{Q_{i}}{\sqrt{Q_{0}^{2}+Q_{1}^{2}+Q_{2}^{2}+Q_{3}^{2}}},i=0,1,2,3$$
$Q_{i}$ represents the output of last fully connected and $q_{i}$ is the q normalization layer output.

The normalization layer strengthens predictive performance. The *q* normalization layer also enhances network training. Two networks were trained: one with and one without q normalization. Excluding normalization, the two networks had the same structure, initialization, and samples. We discovered that the loss corresponded much quicker with q normalization, and the training output was much improved, as shown in Figure 3.

There were two implementation forms of Q-net, full connection implementation and full convolution implementation. Full connection implementation is usually used in a two-stage network to predict the target’s attitude in the Region Proposal Network (RPN). In the full convolution mode, the target’s attitude field was predicted directly in the one-stage network.

### 2.4. Loss Function for Quaternions

Assume that the attitude matrix’s Ground Truth value is $\bar{R}$ and the expected value is *R*. The attitude error measurement equation [3] is described as follows:(4)$$e_{RE}=arccos\left(\frac{Tr\left(\hat{R}{\bar{R}}^{-1}\right)-1}{2}\right)$$

This can prove $Tr({\bar{R}}^{\prime }R)=4{\left(\bar{q}•q\right)}^{2}-1$; see Appendix A.

where *q* and $\bar{q}$ are quaternions, correspond to *R* and $\bar{R}$. So
(5)$$e_{RE}=arccos(\frac{Tr\left({\bar{R}}^{\prime }R\right)-1}{2})=arccos\frac{[4\left(\bar{q}•q{)}^{2}-1\right]-1}{2}=arccos[2\left(\bar{q}•q{)}^{2}-1\right]$$

In ascertaining the disparity among the estimated quaternion q and the real quaternion $\bar{q}$ a loss function needs to be defined. At first, we considered two common loss functions as follows:

L2 loss $|\left|q-\bar{q}\right|{|}_{2}$dot loss $1-<q•\bar{q>}$^2^

However, a quaternion can describe the three-dimensional attitude in the four-dimensional hemispherical space; *q* and −*q* represented the same attitude. For the loss = $|\left|q-\bar{q}\right|{|}_{2}$ or $1-q•\bar{q}$ the maximum error is caused by *q* and −*q*, which could render the estimation unreliable. This is consistent with Equation (4). The symbol of $\bar{q}•q$ has no significance to evaluating the pose similarity of *q* and $\bar{q}$. 

Considering the above problems on common loss functions, we defined a new loss function equation as follows.
(6)$$E_{q}=1-<q,\bar{q}{>}^{2}$$

As the loss function for quaternions, this address the problem where $\bar{q_{i}}$ was the *i*-th component value of ground truth quaternions. Only when the two vectors are exactly the same ($q=\bar{q}$) or reverse $q=-\bar{q}$ does their dot product’s square equal 1. Under the conditions, the minimal value of zero was obtained by Eq Loss. At this time, *q* and $\bar{q}$ described precisely the same attitude, *e_RE_* in Formula (4) catches the minimum value 0.
(7)$$e_{RE}=arccos(\frac{Tr\left({\bar{R}}^{\prime }R\right)-1}{2})=arccos\frac{[4\left(\bar{q}•q{)}^{2}-1\right]-1}{2}=arccos\frac{\left[4\times 1-1\right]-1}{2}=arccos(1)=0$$

The neural network minimizes the loss by continuous optimization of iteration, thus taking the output and ground truth perilously close.

### 2.5. Backward of Quaternions Loss

The learning of unit quaternions was regarded as a regression problem. The output of the network must be a unit vector because of its unique feature of the unit quaternion. According to Equation (3), the network output vector q (Figure 4) was unitized to get Q. The flow chart is as follows:

This kind of unitization was realized by the four feature channels of the output layer. Notably, this is not easy to implement directly through the Graphics Processing Unit (GPU) tensor operation in Python. We derived the backpropagation computation of this unitary processing and enforced it in Darknet.
$$\frac{\partial E_{q}}{\partial Q_{i}}=\sum_{j=0}^{3}\frac{\partial E_{q}}{\partial q_{j}}\times \frac{\partial q_{j}}{\partial Q_{i}}=-2\left(\overset{¯}{q}•q\right)\left[\begin{array}{cccc} \frac{\partial q_{0}}{\partial Q_{0}} & \frac{\partial q_{1}}{\partial Q_{0}} & \frac{\partial q_{2}}{\partial Q_{0}} & \frac{\partial q_{3}}{\partial Q_{0}}\\ \frac{\partial q_{0}}{\partial Q_{1}} & \frac{\partial q_{1}}{\partial Q_{1}} & \frac{\partial q_{2}}{\partial Q_{1}} & \frac{\partial q_{3}}{\partial Q_{1}}\\ \frac{\partial q_{0}}{\partial Q_{2}} & \frac{\partial q_{1}}{\partial Q_{2}} & \frac{\partial q_{2}}{\partial Q_{2}} & \frac{\partial q_{3}}{\partial Q_{2}}\\ \frac{\partial q_{0}}{\partial Q_{3}} & \frac{\partial q_{1}}{\partial Q_{3}} & \frac{\partial q_{2}}{\partial Q_{3}} & \frac{\partial q_{3}}{\partial Q_{3}} \end{array}\right]\left[\begin{array}{c} q_{0}\\ q_{1}\\ q_{2}\\ q_{3} \end{array}\right]$$ and
$$\{\begin{array}{l} \frac{\partial q_{j}}{\partial Q_{i}}=\frac{1}{L}\left(1-\frac{{Q_{i}}^{2}}{L^{2}}\right)=\frac{1}{L}\left(1-{q_{i}}^{2}\right)\;\;i=j\\ \frac{\partial q_{j}}{\partial Q_{i}}=\frac{-Q_{i}Q_{j}}{L^{3}}=\frac{-q_{i}q_{j}}{L}\;\;i\neq j \end{array}$$

The following is the *q* normalization layer’s backward propagation: (8)$$\frac{\partial L_{\mathrm{dot}}}{\partial Q_{i}}=\overset{3}{\underset{j=0}{\sum }}\frac{\partial L_{\mathrm{dot}}}{\partial {\hat{q}}_{j}}\times \frac{\partial {\hat{q}}_{j}}{\partial Q_{i}}=\overset{3}{\underset{j=0}{\sum }}\left(-\frac{1}{2}q_{j}\right)\times \frac{\partial {\hat{q}}_{j}}{\partial Q_{i}}=-\frac{1}{2}\overset{3}{\underset{j=0}{\sum }}q_{j}\times \frac{\partial {\hat{q}}_{j}}{\partial Q_{i}}$$
where
(9)$$\{\begin{array}{l} \frac{\partial {\hat{q}}_{j}}{\partial Q_{i}}=\frac{1}{A}\left(1-\frac{{Q_{i}}^{2}}{A^{2}}\right)=\frac{1}{A}\left(1-{{\hat{q}}_{i}}^{2}\right)\;i=j\\ \frac{\partial {\hat{q}}_{j}}{\partial Q_{i}}=\frac{-Q_{i}Q_{j}}{A^{3}}=\frac{-{\hat{q}}_{i}{\hat{q}}_{j}}{A}\;\;i\neq j\\ A=\sqrt{Q_{0}^{2}+Q_{1}^{2}+Q_{2}^{2}+Q_{3}^{2}} \end{array}$$

### 2.6. Bounding Box to 3D (BB3D)

In this section, an algorithm is proposed to infer the 3D position of the target from the 2D horizontal bounding box and attitude data. Under the condition that it is impossible to obtain the three-dimensional position data of the target for training, it is necessary to quickly infer the object’s three-dimensional position information. BB3D can efficiently perform 3D translation by utilizing the 3D rotation generated by Q-Net along with the 2D bounding box on the original image. The essential concept was to determine 3D translation t using the point-to-side associated constraints. The algorithm was divided into two stages given as follows.

#### 2.6.1. Step 1: The Four Point-to-Side Correspondence Constraints Determination 

On the object’s surface, there are n points, which we named P1, P2, …, Pn, supposing that the object’s coordinate frame’s origin is within the object and that the 3D coordinates of those points are X_1_ = [x1, y1, z1] ^T^, X_2_ = [x2, y2, z2] ^T^, …, X_n_ = [xn, yn, zn] ^T^. The goal here was to decide the four points on the surface that corresponded to the four sides of the 2D box (Figure 5). Two methods were used in this situation.

**Method 1, Indirect comparison:** According to predictions from various viewpoints.

(10)$$z\left[\begin{array}{c} u\\ v\\ 1 \end{array}\right]=KR\left(X-T\right)$$

*K* and *R* represent the camera matrix and rotation matrix, respectively. *u* and *v* represent the pixel coordinate while *z* is the objects’ *z*-coordinate in the camera coordinate frame, which has no effect on the point-to-side correspondence, and it is a positive number.

We inserted the 2D box’s center (*u*_0_, *v*_0_) and the origin of the object coordinate frame, *X*_0_ = (0, 0, 0) *^T^* which was a positive number e.g., *z* = 100, into Equation (10).
(11)$$z\left[\begin{array}{c} u_{0}\\ v_{0}\\ 1 \end{array}\right]=KR\left(X_{0}-T\right)$$

We get
(12)$$T_{0}=-z{\left(KR\right)}^{-1}\left[\begin{array}{c} u_{0}\\ v_{0}\\ 1 \end{array}\right]$$

*T*_0_ is not a true 3D translation, but it can assist in measuring the point-to-side correspondence. We took *z*, *T*_0_ into Equation (10) to get Equation (13).
(13)$$z\left[\begin{array}{c} u\\ v\\ 1 \end{array}\right]=KR\left(X-T_{0}\right)$$

After, we used Equation (13) to compute n pairs of pixel coordinates, (*u*1, *v*1), (*u*2, *v*2), …, (*un*, *vn*). The max *u*, *v*, and min *u*, *v* were found amongst them. The right side of the 2D box was referred to by the point that resulted in max *u*; we recorded its index and called it iR. Likewise, when *u* touched the left side, the point appeared; we recorded its index and named it iL. We reported the index and named it iB because the point that results in max *v* reached the bottom side. We registered its index and named it iT because the point results in min *v* touched the upside. With this, we established four points that correspond to the two-dimensional box’s four sides.

**Method 2 Direct conversion:** In the case of the n points on the surface,

(14)$$z_{i}\left[\begin{array}{c} u_{i}\\ v_{i}\\ 1 \end{array}\right]=KR\left(X_{i}-T_{0}\right)$$

Taking (12) into (14),
(15)$$z_{i}\left[\begin{array}{c} u_{i}\\ v_{i}\\ 1 \end{array}\right]=KR\left(X_{i}+z{\left(KR\right)}^{-1}\left[\begin{array}{c} u_{0}\\ v_{0}\\ 1 \end{array}\right]\right)=KRX_{i}+z\left[\begin{array}{c} u_{0}\\ v_{0}\\ 1 \end{array}\right]$$

We created a coordinate frame with the origin at the object’s center, and three axes were corresponding to the camera’s three axes, respectively. Pi’s coordinates are *RX_i_* = [∆*Xi*, ∆*Yi*, ∆*Zi*] in this coordinate frame, referring to the coordinate transformation principle. This produced the following:(16)$$KRX_{i}=\left[\begin{array}{c} f_{x}\Delta X_{i}+c_{x}\Delta Z_{i}\\ f_{y}\Delta Y_{i}+c_{y}\Delta Z_{i}\\ \Delta Z_{i} \end{array}\right]=\left[\begin{array}{c} P_{xi}\\ P_{yi}\\ \Delta Z_{i} \end{array}\right]$$

The components of the camera matrix are *f_x_*, *f_y_*, *c_x_*, *c_y_*. *z* is the z-coordinate of the object in the camera coordinate frame, and it is much larger than $\Delta Z_{i}$. So
(17)$$\frac{\Delta Z_{i}}{z}\approx 0$$

From Equations (15)–(17) we get
(18)$$\{\begin{array}{l} u_{i}=\frac{f_{x}\Delta X_{i}+c_{x}\Delta Z_{i}+zu_{0}}{\Delta Z_{i}+z}=\frac{\frac{f_{x}\Delta X_{i}}{z}+\frac{c_{x}\Delta Z_{i}}{z}+u_{0}}{\frac{\Delta Z_{i}}{z}+1}\approx \frac{f_{x}\Delta X_{i}}{z}+u_{0}\\ v_{i}=\frac{f_{y}\Delta Y_{i}+c_{y}\Delta Z_{i}+zv_{0}}{\Delta Z_{i}+z}=\frac{\frac{f_{y}\Delta Y_{i}}{z}+\frac{c_{y}\Delta Z_{i}}{z}+v_{0}}{\frac{\Delta Z_{i}}{z}+1}\approx \frac{f_{y}\Delta Y_{i}}{z}+v_{0} \end{array}$$

From (18), we can see that the values of *u_i_* and *v_i_* are determined by $\Delta X_{i}$ and $\Delta Y_{i}$.If a positive number is assigned to z then the following simplifications occur in Equation (16):(19)$$RX_{i}=\left[\begin{array}{c} \Delta X_{i}\\ \Delta Y_{i}\\ \Delta Z_{i} \end{array}\right]$$

We entered the 3D coordinates of n points into Equation (19) and calculated *n* pairs of [∆*X*, ∆*Y*, ∆*Z*], to infer four points index iL, iR, iT, iB that corresponded to the max and min ∆*X*, ∆*Y*, and calculated the four points corresponding to the four sides of the 2D bounding box.

#### 2.6.2. Step 2: 3D Translation Determination

The four points were the outcomes of the first step and their indexes were named iL, iR, iT, and iB. According to the collinearity equation,
(20)$$\{\begin{array}{l} u_{Ki}=\frac{u_{i}{-c}_{x}}{f}=\frac{r_{11}\left(x_{i}-t_{x}\right)+r_{12}\left(y_{i}-t_{y}\right)+r_{13}\left(z_{i}-t_{z}\right)}{r_{31}\left(x_{i}-t_{x}\right)+r_{32}\left(y_{i}-t_{y}\right)+r_{33}\left(z_{i}-t_{z}\right)}\\ v_{Ki}=\frac{v_{i}{-c}_{y}}{f}=\frac{r_{21}\left(x_{i}-t_{x}\right)+r_{22}\left(y_{i}-t_{y}\right)+r_{23}\left(z_{i}-t_{z}\right)}{r_{31}\left(x_{i}-t_{x}\right)+r_{32}\left(y_{i}-t_{y}\right)+r_{33}\left(z_{i}-t_{z}\right)} \end{array}$$

*T* is the camera’s 3D coordinate in the object coordinate frame, where (*t_x_ t_y_ t_z_*) are the elements of *T* and unknown. (*x_i_, y_i_, z_i_*) are the three-dimensional coordinates. We get (21) after linearizing (20).
(21)$$\{\begin{array}{l} \left(u_{Ki}r_{31}-r_{11}\right)t_{x}+\left(u_{Ki}r_{32}-r_{12}\right)t_{y}+\left(u_{Ki}r_{33}-r_{13}\right)t_{z}=\left(u_{Ki}r_{31}-r_{11}\right)x_{i}+\left(u_{Ki}r_{32}-r_{12}\right)y_{i}+\left(u_{Ki}r_{33}-r_{13}\right)z_{i}\\ \left(v_{Ki}r_{31}-r_{21}\right)t_{x}+\left(v_{Ki}r_{32}-r_{22}\right)t_{y}+\left(v_{Ki}r_{33}-r_{23}\right)t_{z}=\left(v_{Ki}r_{31}-r_{21}\right)x_{i}+\left(v_{Ki}r_{32}-r_{22}\right)y_{i}+\left(v_{Ki}r_{33}-r_{23}\right)z_{i} \end{array}$$

In view of the four point-to-side correspondence constraints, we get
(22)$$\{\begin{array}{l} \left(u_{KiL}r_{31}-r_{11}\right)t_{x}+\left(u_{KiL}r_{32}-r_{12}\right)t_{y}+\left(u_{KiL}r_{33}-r_{13}\right)t_{z}=\left(u_{KiL}r_{31}-r_{11}\right)x_{iL}+\left(u_{KiL}r_{32}-r_{12}\right)y_{iL}+\left(u_{KiL}r_{33}-r_{13}\right)z_{iL}\\ \left(u_{KiR}r_{31}-r_{11}\right)t_{x}+\left(u_{KiR}r_{32}-r_{12}\right)t_{y}+\left(u_{KiR}r_{33}-r_{13}\right)t_{z}=\left(u_{KiR}r_{31}-r_{11}\right)x_{iR}+\left(u_{KiR}r_{32}-r_{12}\right)y_{iR}+\left(u_{KiR}r_{33}-r_{13}\right)z_{iR}\\ \left(v_{KiT}r_{31}-r_{21}\right)t_{x}+\left(v_{KiT}r_{32}-r_{22}\right)t_{y}+\left(v_{KiT}r_{33}-r_{23}\right)t_{z}=\left(v_{KiT}r_{31}-r_{21}\right)x_{iT}+\left(u_{KiT}r_{32}-r_{22}\right)y_{iT}+\left(u_{KiT}r_{33}-r_{23}\right)z_{iT}\\ \left(v_{KiB}r_{31}-r_{21}\right)t_{x}+\left(u_{KiB}r_{32}-r_{22}\right)t_{y}+\left(v_{KiB}r_{33}-r_{23}\right)t_{z}=\left(v_{KiB}r_{31}-r_{21}\right)x_{iB}+\left(u_{KiB}r_{32}-r_{22}\right)y_{iB}+\left(u_{KiB}r_{33}-r_{23}\right)z_{iB} \end{array}$$(22) can be stated in matrix form, as shown below:(23)$$AT=X_{box}$$
where
(24)$$A=\left[\begin{array}{ccc} u_{KiL}r_{31}-r_{11} & u_{KiL}r_{32}-r_{12} & u_{KiL}r_{33}-r_{13}\\ u_{KiR}r_{31}-r_{11} & u_{KiR}r_{32}-r_{12} & u_{KiR}r_{33}-r_{13}\\ v_{KiT}r_{31}-r_{21} & v_{KiT}r_{32}-r_{22} & v_{KiT}r_{33}-r_{23}\\ v_{KiB}r_{31}-r_{21} & u_{KiB}r_{32}-r_{22} & v_{KiB}r_{33}-r_{23} \end{array}\right]=\left[\begin{array}{c} b_{Left}\\ b_{Right}\\ b_{Top}\\ b_{Bottom} \end{array}\right]$$

Bounding Box Equation (23), by putting Equation (24) where the matrix *A* is the Bounding Box Matrix, and the four-row vectors Side Vector, gave Equation (25).


(25)$$\{\begin{array}{l} u_{KiL}=\frac{x_{L}{-c}_{x}}{f_{x}}\\ u_{KiR}=\frac{x_{R}{-c}_{x}}{f_{y}}\\ v_{KiT}=\frac{y_{T}{-c}_{y}}{f_{x}}\\ v_{KiB}=\frac{y_{B}{-c}_{y}}{f_{y}} \end{array}$$


*X_L_*, *Y_L_*, *X_R_*, *Y_R_* are the pixel coordinates of the 2D bounding box’s borders.
(26)$$X_{box}=\left[\begin{array}{c} \left(u_{KiL}r_{31}-r_{11}\right)x_{iL}+\left(u_{KiL}r_{32}-r_{12}\right)y_{iL}+\left(u_{KiL}r_{33}-r_{13}\right)z_{iL}\\ \left(u_{KiR}r_{31}-r_{11}\right)x_{iR}+\left(u_{KiR}r_{32}-r_{12}\right)y_{iR}+\left(u_{KiR}r_{33}-r_{13}\right)z_{iR}\\ \left(v_{KiT}r_{31}-r_{21}\right)x_{iT}+\left(u_{KiT}r_{32}-r_{22}\right)y_{iT}+\left(u_{KiT}r_{33}-r_{23}\right)z_{iT}\\ \left(v_{KiB}r_{31}-r_{21}\right)x_{iB}+\left(u_{KiB}r_{32}-r_{22}\right)y_{iB}+\left(u_{KiB}r_{33}-r_{23}\right)z_{iB} \end{array}\right]=\left[\begin{array}{c} b_{Left}•X_{iL}\\ b_{Right}•X_{iR}\\ b_{Top}•X_{iT}\\ b_{Bottom}•X_{iB} \end{array}\right]$$

After calling four-row vectors Bounding Box Vector, the norm of Side Vector is given as follows:(27)$$\{\begin{array}{l} \left||b_{Left}{\left|\right|}^{2}=\right||\left\{u_{KiL}r_{31}-r_{11},u_{KiL}r_{32}-r_{12},u_{KiL}r_{33}-r_{13}\right\}{\left|\right|}^{2}=u_{KiL}^{2}+1\\ \left||b_{Right}{\left|\right|}^{2}=\right||\left\{u_{KiR}r_{31}-r_{11},u_{KiR}r_{32}-r_{12},u_{KiR}r_{33}-r_{13}\right\}{\left|\right|}^{2}=u_{KiR}^{2}+1\\ \left||b_{Top}{\left|\right|}^{2}=\right||\left\{v_{KiT}r_{31}-r_{21},v_{KiT}r_{32}-r_{22},v_{KiT}r_{33}-r_{23}\right\}{\left|\right|}^{2}=v_{KiT}^{2}+1\\ \left||b_{Bottom}{\left|\right|}^{2}=\right||\left\{v_{KiB}r_{31}-r_{21},v_{KiB}r_{32}-r_{22},v_{KiB}r_{33}-r_{23}\right\}{\left|\right|}^{2}=v_{KiB}^{2}+1 \end{array}$$

To solve the Bounding Box Equation, we employed the least-squares technique.
(28)$$T={\left(AA^{T}\right)}^{-1}A^{T}X_{box}$$

*T* is the camera’s 3D coordinate in the object coordinate frame. 

3D translation *t*, as the object’s 3D coordinate in the camera coordinate, was obtained by the follow equation:
*T* = −*RT*(29)


## 3. Results

### 3.1. Experiments on LineMod

#### 3.1.1. Implementation Details

We used the LineMod dataset [9] to test our method in this section. On each image, there was only one study object. To obtain the 2D bounding box, we used Multi-task Cascaded Convolutional Networks (MTCNN) [32] as the 2D detection process. The 6D pose was then calculated using Q-Net and the Bounding Box Equation.

We used a mini-batch size of 24 images and variable learning rates to train Q-Net by Stochastic gradient descent (SGD) with momentum, with “step” as the learning rate policy. The gamma was set to 0.8 and the step size was set to 104. The following values were held constant: momentum 0.9, weight decay 4 × 10^−3^, and base learning rate 10^−4^. For weights and zero-initialize biases, Xavier initialization was selected. Dropout was used behind the fully connected layer. We trained and tested all of the models with Caffe.

In the prediction phase, we first used MTCNN to get the 2D rectangle, then used Q-net to get the target’s pose in each rectangle, and finally used BB3D to get the 3D coordinates. 

We used 200 validation images to evaluate the projection point error, attitude, and position data error of 13 kinds of objects.

#### 3.1.2. Visualization Results

The results of the 6D pose estimate on LineMod are shown in Figure 6. We used all of the 3D coordinates from the Bounding Box Equation point cloud, as LineMod had a point cloud for each object.

#### 3.1.3. Performance Evaluation

Pixel projection error is an intuitive index to evaluate the prediction accuracy of a 6D Pose. We used the validation dataset to evaluate the pixel projection error. The data given below in Table 1 describes the comparison of our method with SSD, BB8, and other methods:

According to Hodăn [3], the translational error (*e_TE_*) and rotational error (*e_RE_*) are as follows:(30)$$e_{RE}\left(\hat{R},\bar{R}\right)=arccos\left(\left(Tr\left(\hat{R}{\bar{R}}^{-1}\right)-1\right)/2\right)$$
(31)$$e_{TE}\left(\hat{t},\bar{t}\right)=||\bar{t}-\hat{t}|{|}_{2}$$

We further obtained the position and attitude errors of the objects in the LineMod as follows, given in Table 2.

Table 1 displays the pixel projection errors of 13 different types of objects in the LineMod. Table 2 has shown the 6D Pose error *e_TE_* and *e_RE_* results for 13 objects, as well as method comparisons. Both the 2D Project and 6D Pose accuracy assessment requirements from the above two tables indicate that this algorithm outperformed BB8 and SS6D.

The advantages of our method are due to the following three reasons:Both BB8 and SSD were trained based on the pixel projection error, and the same length of pixel error has an entirely different influence on the attitude error of the small-scale object and the large-scale object. The biggest problem of traditional projection errors was that they could not balance the weight between small-scale objects and large-scale objects. Our method avoided this problem.Q-net uses a more reasonable quaternion to predict attitude and a more reasonable loss object function: Equation = 1−< q, q’ >^2^.Bb8 and SS6D only consider the projection of the eight corners of the rectangular cube of the target, while our BB3D considers the projection constraint of the n >8 points of the 3D model of the object in 3D space.

To evaluate Eq’s effect as a loss function, we compared the prediction accuracy of Equation (5) loss function in Section 2.4. with quaternion with L2 loss and Euler angle model in Table 3.

It can be seen from Table 3 that Equation (6) defined in Section 2.4 can achieve higher attitude accuracy than L2 q loss. This is because the maximum error was generated by q and –q, rendering the prediction unstable. We solved this problem by defining a new loss function, $E_{q}=1-<q,\bar{q}{>}^{2}$.

Accuracy was also higher than the accuracy of the direct prediction of the Euler angle. When the object attitude was regressed through three Euler angles, the problem of angle cycle was encountered; that is, θ and θ + 2π described the same angle, constituting the most significant L2 loss.

To further obtain the location error distribution of LineMod predicted by BB3D, we have shown the histogram of the translation error distribution between the predicted value of LineMod and ground truth in Figure 7.

It can be seen that the distribution of translation error distribution obtained by BB3D is around 0, which is effective for the prediction of the target location.

It can be seen that the error distribution of BB3D prediction is mostly concentrated in the interval. Table 4 shows the average time consumption of BB3D. These experimental data have shown that the BB3D algorithm was easy to implement, fast, and met the needs of fast target positioning under the condition of being unable to obtain the object’s three-dimensional position data for training.

There are 8^4^ = 4096 possible configurations, since each side of the 2D detection box can react to any of the eight corners of the 3D box. The referenced method [28], 3D Bounding Box, needed to solve the equation 4096 times to get the correct solution, which could not be realized in the ordinary front-end system at all. However, our BB3D only needed to solve the equation one time, and the speed was increased by a thousand times.

### 3.2. Computation Times

Table 5 shows the comparison of our Q-Net with SS6D [24] and BB8 [27] on speed. All the speeds are tested on LineMod dataset images with the size of 640 × 480 on a GeForce GTX 2080 GPU, Intel Core i7-5820K 3.30 GHz. The time of Q-Net inference and BB3D was included.

It can be seen from Table 5 that the time consumption of our Q-net + BB3D method is far less than that of the classical methods SS6D and BB8, and it can run stably at 6.88 ms/frame in real-time.

### 3.3. Experimental Results on the KITTI Dataset

In order to better verify the effectiveness of Q-net attitude prediction, we also performed attitude learning on the KITTI dataset. 481 pieces were randomly selected from the KITTI dataset for verification, and the remaining 7000 pieces were used for training. We converted the attitude in the KITTI data set into quaternion, used Q-net to predict the attitude quaternion, and used the formula Eq = 1−<q, q’>^2^ as a loss function.

Our data source is the 2D image without direct depth information. However, the 3D pose of the object can be predicted by the model constructed by the visual knowledge of the deep neural network. The model used the YOLO convolution network to predict the 2D bounding box of the object, and the model input size was 3 × [832 × 416]. It supported the following 8 categories of object 3D detection in KITTI data: truck, car, cyclist, misc, van, tram, person sitting. Q-net was used to predict the attitude quaternion of the object. Finally, according to the predicted 3D pose and position, 8 vertices of the object’s outer cube were projected onto the original image, and the visualization effect is as shown in Figure 8.

In Figure 8, the red arrow is the *X*-axis of the object ontology, the green arrow is the *Y*-axis of the object ontology, and the blue arrow is the *Z*-axis of the object ontology. It can be seen from the above figure that the pose determined by the *XYZ* three axis of the object conforms to the expectation of human vision for 3D scene understanding.

After 107 epoch iterations, several technical indexes related to the 3D pose of the model are as follows, given in Table 6.

It can be seen from the above table that the Q-net method introduced in this paper can also achieve good results on the KITTI dataset.

## 4. Discussion

It is very important be able to get the pose of an object according to an image. In the field of automatic driving, we can get the direction and distance of vehicles in front of each other in relation to the camera. In the field of remote sensing, we can get the movement direction and trend of the ground target. Traditional feature-based methods and template-based methods can achieve certain results. However, the adaptability of these technologies to big data is limited. For example, the RGB-D method uses depth data to predict the object, but this method cannot directly get the position and pose information of an object, and the active depth sensor consumes a lot of power, which requires additional hardware costs. Therefore, we chose quaternion q as our 3D rotation representation. We propose a new and effective quaternion loss function to solve the problem of pose difference evaluation by L2loss. On this basis, a boundary box equation algorithm BB3D is proposed to solve the three-dimensional translation according to the 2D bounding box of the object, which greatly improves the calculation speed.

## 5. Conclusions

Based on the 2D bounding box, this study proposed a method for obtaining a 6D pose from a single RGB image. This method closely resembled the previous 2D object detection algorithm. We used the Bounding Box Equation to obtain a 3D translation after training Q-Net on Dot Product Loss for regression of the unit quaternion. Experiments have shown that the strategy is appropriate, efficient, and feasible. Until now, our approach has been limited to estimating the pose of a single type of entity. We aim to boost productivity and focus on extending this method to solve multi-category object pose estimation. We also intend to extend the dataset to include daily commonplace objects.

## Figures and Tables

**Figure 1 sensors-21-02939-f001:**
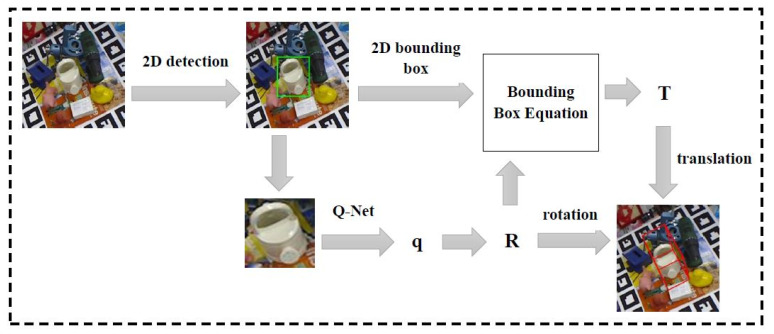
The Flow diagram of our approach.

**Figure 2 sensors-21-02939-f002:**
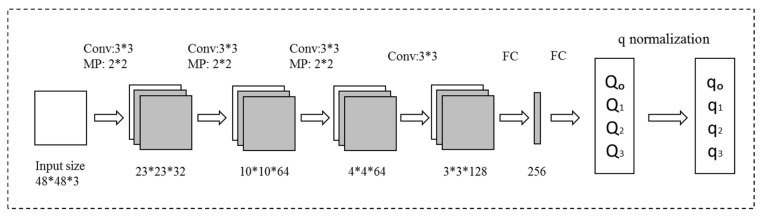
The architecture of Q-Net. “Conv” stands for convolution, “MP” stands for max pooling, and “FC” stands for fully connected. Convolution and max-pooling have step sizes of 1 and 2, respectively.

**Figure 3 sensors-21-02939-f003:**
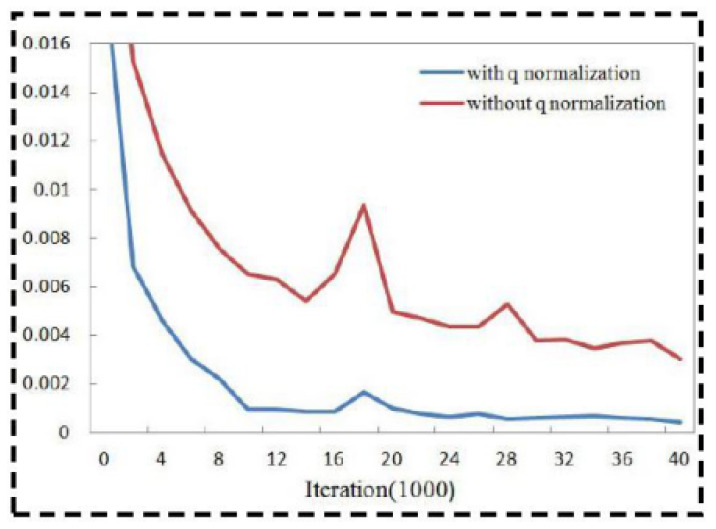
The value of the loss is expressed by the ordinate axis, while the iteration is represented by the abscissa axis.

**Figure 4 sensors-21-02939-f004:**
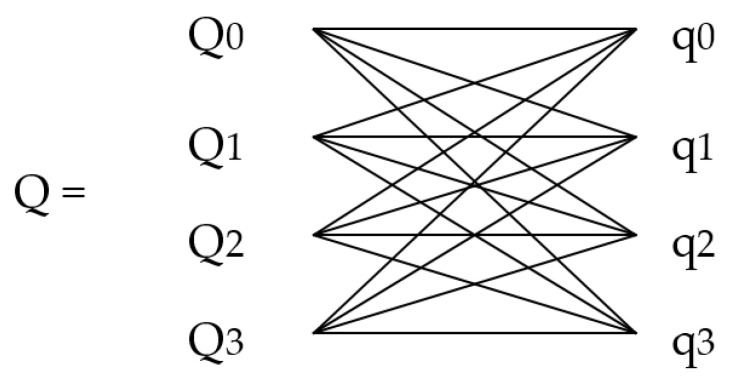
The constraint layer transforms the outputs of the four neurons {Q_0_, Q_1_, Q_2_, Q_3_} into quaternion unitized vectors.

**Figure 5 sensors-21-02939-f005:**
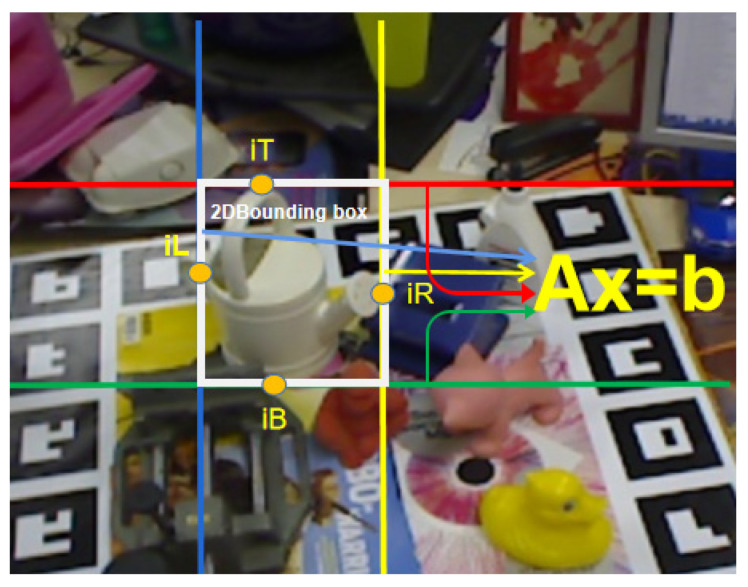
The point-to-side correspondence. The object’s surface was made up of a large number of points. The four sides were assumed to be touched by four-point indexes iL, iR, iT, and iB.

**Figure 6 sensors-21-02939-f006:**
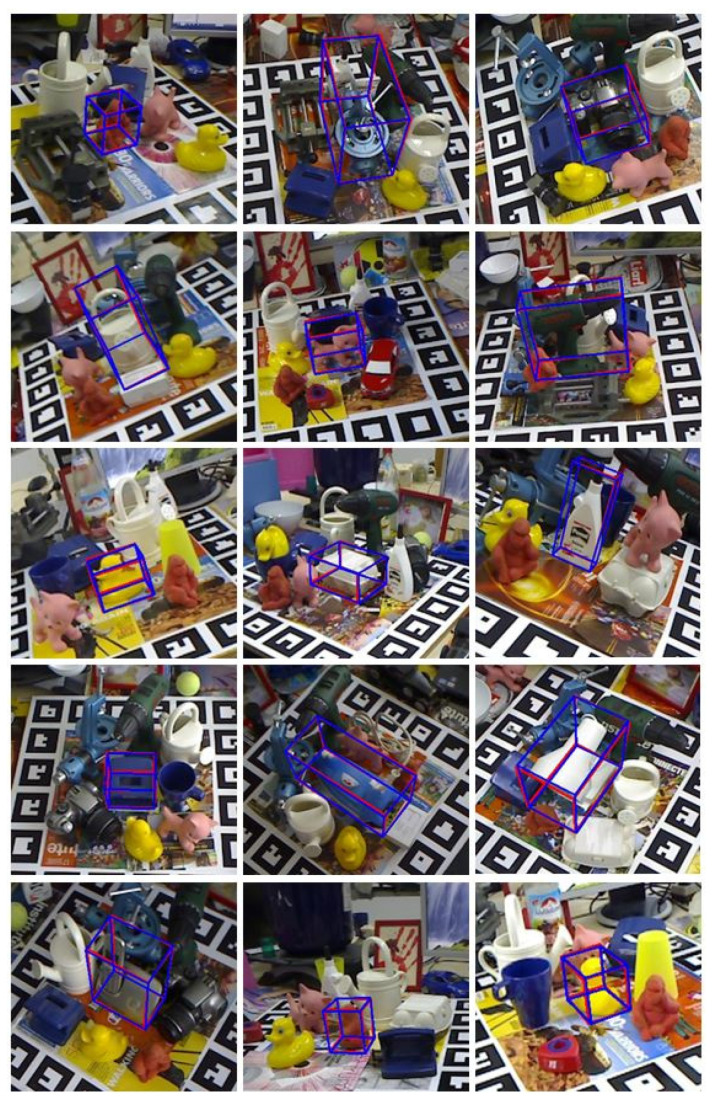
Our method’s results of pose estimation on the LineMod dataset. The ground truth is represented by the blue 3D bounding boxes, while the estimation results are represented by the red 3D bounding boxes.

**Figure 7 sensors-21-02939-f007:**
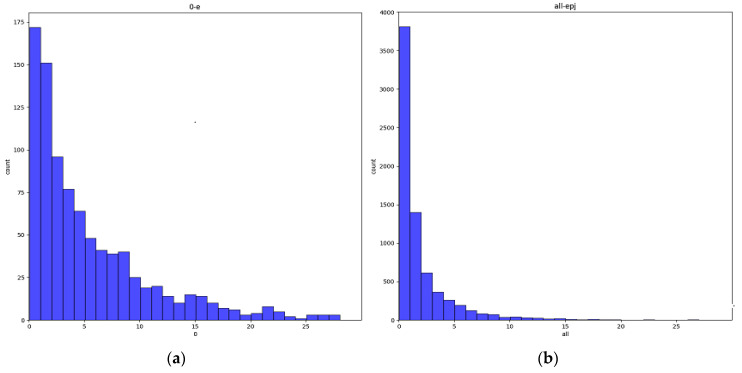
Error distribution of BB3D prediction (**a**) Histogram of the translation error distribution (**b**) Histogram of the pixel projection error distribution.

**Figure 8 sensors-21-02939-f008:**
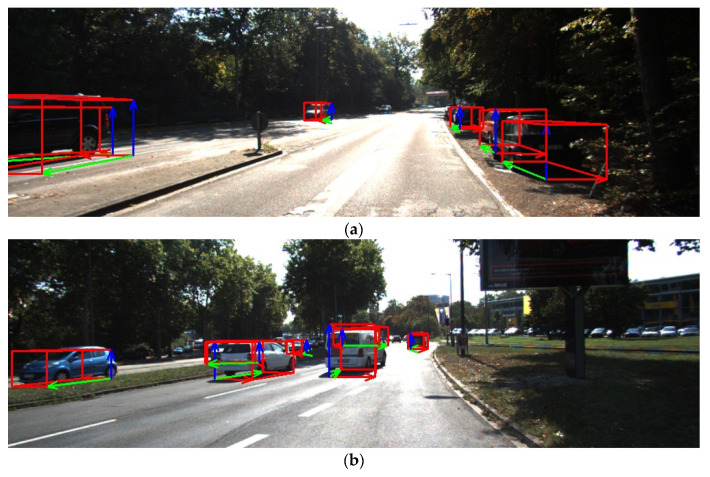
Visualization of attitude prediction by Q-net running on KITTI dataset.

**Table 1 sensors-21-02939-t001:** LineMod 2 objects of 2D Projection pixel and projection error comparison.

Object Class	5 Pixels Accuracy				Averaged Pixel Projection Error (Pixels)
	Our Method	SS6D [24]	BB8 [27]	Euler Angle	Our Method
Ape	0.9894	0.9210	0.9530	0.912	1.980852
Cam	0.9658	0.9324	0.809	0.431	2.642408
Glue	0.9680	0.9653	0.890	0.693	2.673251
Box	0.9457	0.9033	0.879	0.682	2.543196
Can	0.9130	0.9744	0.841	0.625	3.171883
Lamp	0.9347	0.7687	0.744	0.504	2.506023
bench	0.7152	0.9506	0.800	0.802	4.251370
Cat	0.9826	0.9741	0.970	0.931	2.534734
hole	0.9352	0.9286	0.905	0.782	2.613619
Duck	0.9534	0.9465	0.812	0.679	2.583827
Iron	0.9015	0.8294	0.789	0.645	2.522091
Driller	0.8985	0.7941	0.7941	0.465	2.604456
phone	0.9458	0.8607	0.776	0.469	2.698741
avg	0.926831	0.9037	0.839	0.663077	2.717419
bowl	0.956204	-	-		2.672496
Cup	0.932546	-	-		2.987507

**Table 2 sensors-21-02939-t002:** Comparison of *e_TE_* and *e_RE_* results between our method, SS6D, and BB8 for 13 objects.

Object Class	*e_RE_* (Degrees)			*e_TE_* (cm)		
	Our Method	BB8 [27]	SS6D [24]	Our Method	BB8 [27]	SS6D [24]
Ape	2.451113	2.946983	2.75364	1.865379	1.854367	1.867834
Cam	2.412643	2.550849	2.82091	1.842576	1.897132	1.792546
Glue	2.512097	2.382351	2.63785	1.632802	1.980421	1.872435
Box	2.103465	2.400987	2.56307	1.543097	1.784109	1.637091
Can	2.281664	2.139032	2.89345	1.809013	1.976015	1.873506
Lamp	2.26914	2.109743	2.76324	1.506136	1.679213	1.612308
Bench	2.45763	2.834509	2.81533	1.637125	1.789204	1.738479
Cat	2.25131	2.436078	2.89530	1.534072	1.563078	1.853047
Hole	2.314566	2.765301	2.91036	1.613631	1.659056	1.635032
Duck	2.471108	2.535874	2.74382	1.583450	1.723480	1.710795
Iron	2.537642	2.540321	2.62139	1.522011	1.553201	1.542103
Driller	2.344567	2.395455	2.65937	1.594294	1.663067	1.653411
Phone	2.290789	2.415276	2.673854	1.698068	1.703409	1.699508
Average	2.3613642	2.496366	2.750122	1.644743	1.755827	1.729853

**Table 3 sensors-21-02939-t003:** Comparison of angle error *e_RE_* with the different loss functions.

Object Class	Angle Error *e_RE_*		
	Quaternion with Eq Loss	Quaternion with L2 Loss	Euler Angle Model
Ape	2.451113	2.73248	2.68213
Cam	2.412643	2.76312	2.79362
Glue	2.512097	2.80914	2.84532
box	2.103465	2.64539	2.39451
Can	2.281664	2.71356	2.50638
Lamp	2.26914	2.59823	2.43572
bench	2.45763	2.67291	2.7918
Cat	2.25131	2.63078	2.58342
hole	2.314566	2.62335	2.70135
Duck	2.471108	2.71123	2.70816
Iron	2.537642	2.86204	2.88425
Driller	2.344567	2.63417	2.67134
phone	2.290789	2.58392	2.58239
average	2.3613642	2.690794	2.66003

**Table 4 sensors-21-02939-t004:** Time consumption of bb3d for Q-Net.

	Cost Time 1000 Instances (MS)	Average Cost Time (MS)
BB3D (*n* points)	34,000	34
BB3D (8 points)	1273	1.273
Algorithm in [28] (8 points)	3,786,923	3786.923

**Table 5 sensors-21-02939-t005:** Cost time comparison for LineMod dataset.

Method	Cost Time for One Object (MS)		
SS6D [24]	18.67		
BB8 [27]	91.45		
	Q-Net	BB3D	total
Q-Net	5.68	1.2	6.88

**Table 6 sensors-21-02939-t006:** The technical index of the pose in model training.

Model Technical Index	Value	Description	Changes in the Training Process
Avgerage_loss	0.882620	The overall loss in the training process.	In the training process, loss decreases with the increase of iteration times, which indicates that the learning process of the model can converge
IOU	0.811207	The average value of IOU between the predicted box and the ground truth box.	In the training process, with the increase of iterations, the IOU is increasing, which indicates that the object box prediction is gradually accurate
Pose_dot	0.974438	The dot product between Q-net predicted object attitude quaternion and true value attitude quaternion	In the training process, with the increase of the number of iterations, the value of poses dot increases and Approaches 1. This indicates that the quaternion of object attitude obtained by Q-net is approaching the true value.

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
