# Peer review of "Estimation of 6D Object Pose Using a 2D Bounding Box"

_sensors, 2021, doi:10.3390/s21092939_

Round 1

Reviewer 1 Report

This must seem like a disappointing review.  I did not take me long to realise that I was reviewing a paper which is a reworking of Liu and He (2019).  Both papers claim novelty and infer new knowledge.  Yet,  the publications are three years apart.  Perhaps the tables on pages 13 to 15 are more detailed than in Liu and He (2019).

If this paper is a development of the 2019 study then the paper should clearly state this, explain the substantive differences in the two studies and expand the paper considerably with more emphasis on new knowledge and less on the original work except for appropriate references.   That, said, I doubt if there is enough to warrant an entire paper.

The 2019 paper is not cited or any of its contents and yet substantial sections of it are re-produced in this study.  Some effort has been made at rewording and moving things around to some extent.  Technically, without appropriate citation of the 2019 paper, this could be considered plagiarism.  

I have not written the paper off for the moment so that the authors can consider my comments, assess their validity and, if they so wish, to  resubmit a substantively new paper.

My response of Not applicable in Recommendations for Authors merely reflects that the work has already been done it seems to me.  If, for example, the work has been repeated with similar precision for example, this would strengthen the rigour of the work.

I trust these comments are helpful.

New comments:

I recommend that the paper 'Estimation of 6D Object Pose Using a 2D Bounding Box' be published subject to the following minor amendments,

There are several instances in which the grammar should be improved and/or reworded for clarity, for example,

1/39 (page number/line number) 'However, these modalities have a legal restriction that appropriate textures on the objects are required'. Please clarify/reword. 

2/48 'RGB-D methods profound cameras prevail'. Meaning, please clarify? 

2/58-59 Please reword for clarity.

2/68 By PnP it is assumed you are referring to Point n Perspective?  State the full descriptor and then the abbreviation just for clarity and do so for the first mention of other abbreviations such as BB8 RPN and so on so that your abbreviations are consistent.

2/78 What is meant by 'expense the computational method'? 

2/82-83 'We have developed an entirely new way of assessing the 3D rotation R and 3D translation rotation t in only one RGB image'.  Emphasise this in the abstract to capture the readers' interest and demonstrate the novelty of the research at an early stage in the paper.

3/Figure 1 Perhaps 'Flow diagram of our approach' or something similar, would be a more appropriate caption. 

11/305 Figure 6 is, apart from a different caption, similar to Figure 2 and is not discussed in the text.  It is redundant therefore and should be omitted.

1/17-19 'We evaluated our model using the LineMod dataset, and experiments have shown that our methodology is more acceptable and efficient'. Could you spell out what you mean by more acceptable and efficient?  Than what, for example? This does not need to be longwinded, one or two sentences with examples would suffice.

Elements of the paper bear a striking resemblance to Park et al., 2019 (paper attached), is the paper an improvement on Park et al., or does it set out an alternative approach perhaps?  The object set up for example in Figure 1 is similar to Park et al., Figure 3, if there is a reason for this then it should be stated.  Is it a standard set up for example in which similar objects are used?  Please clarify as this has implications for the methodology.

There are other very minor issues such as 2/53 (remove superfluous full stop mid-sentence). The paper would benefit from another thorough read through to clear up any other minor issues.  This would enhance the professionalism of the presentation.

Author Response

Kindly find the attached file. All the answers and modifications (suggested) are mentioned in the file.

Reviewer 2 Report

The paper proposes an interesting method for 6D pose estimation of objects using 2D bounding boxes. A new quaternion pose loss function and Q-Net are developed to estimate the object pose. Bounding Box Equation has been proposed to obtain 3D translation from 2D bounding boxes. Extensive evaluation of the proposed technique has been performed on LineMod dataset. Empirical results look promising.

I have few comments:

While the paper introduces an interesting technique, its motivation and technical/practical significance are not clear. It would be good to have this information in the paper. Please clearly state that in the introduction.

As it is easy to capture RGB-D images, why did the authors use RGB images only in their experiment?

The proposed technique has been tested only on one dataset. It is difficult to assess the effectiveness and robustness of this technique.

What's the performance of the proposed technique in cluttered scenes with fully or partially occluded objects?

Block diagrams are blurry. Please include good quality figures in the manuscript.

Author Response

Thank you for consideration and suggestions. kindly find the attached file for detailed answers of the comments.

Reviewer 3 Report

The submitted paper proposes an effective method for detecting or estimating the 6Dimentional (6D) pose of objects from an RGB image. The authors evaluated the proposed model using the LineMod dataset, and experiments have shown that our methodology is more acceptable and efficient.

There are some comments:
1. English representation needs to be improved.

2. Page 3, Line98: you say you proposed a new unitization layer, what is the new unitization layer. It’s the normalized layer. I can't think of any connection between them.

3. Page 4, Figure 2. Why would you use 2*2 on the last conv.

4. Page 4, Figure 2. The Q-NET is very similar to the network of the paper “Pix2Pose: Pixel-Wise Coordinate Regression of Objects for 6D Pose Estimation”.

5. Page 6, Line172. Formula nonstandard

6. Page 11, Figure 6, this picture appears the second time, and if you want to talk about this network you should put it up front and not the results here.

7. The author should include the following archival references:
Zhang D, He L, Tu Z, et al. Learning motion representation for real-time spatio-temporal action localization[J]. Pattern Recognition, 2020, 103: 107312. 

Author Response

Thank you for your time and suggestions. Kindly, find the attached file for details. 

Round 2

Reviewer 1 Report

Thanks to the authors for addressing my comments.

Just a few very minor amendments to add a bit of polish to the final paper as indicated on the attached pdf.  No need to resubmit for further review.

Best wishes.

Author Response

Thank you for your valuable suggestions. We have modified our manuscript accordingly. 

Reviewer 2 Report

Thanks for addressing my comments from the first revision round.

Could you please report the results for Kitti dataset in the paper?

RGBD scanners are widely used in research and industry for indoor applications and have produced very good results. They are cheap and portable. I am not sure what do authors mean by "bulky and expensive". This requires clarification.

In regards to authors' response, "In practical applications, we are more concerned about the three-dimensional position and posture of the target relative to the camera, which will be more convenient for us to understand the semantic information of the target in the scene."

You are not dealing with 3D data. The paper estimates 6D pose from a 2D bounding box. This requires a bit of clarification.

Author Response

Thank you for your suggestions and comments. We have modified our manuscript accordingly. Kindly find the attached file for more details.
